

# Hand-written letters and photo albums linking geoscientists with school classes

Mathew A. Stiller-Reeve[1,2], Claudio Argentino[3], Kate Alyse Waghorn[3], Sunil Vadakkepuliyambatta[3,4], Dimitri Kalenitchenko[3,5,6], Giuliana Panieri[3]

[1]Konsulent Stiller-Reeve, 5281 Valestrandsfossen, Norway
[2]University of Bergen, Centre for Climate and Energy Transformation (CET), Faculty of Social Sciences, PO Box 7802, 5020 Bergen, Norway
[3]CAGE, Centre for Arctic Gas Hydrate, Environment and Climate, UiT, The Arctic University of Norway, Tromsø, 9010, Norway
[4] National Centre for Polar and Ocean Research, Ministry of Earth Sciences, Vasco da Gama, Goa, India
[5] LIttoral ENvironnement et Sociétés (LIENSs)—UMR 7266, La Rochelle, France
[6] Department of Arctic and Marine Biology, The Arctic University of Norway, Tromsø, Norway

*Correspondence to*: Mathew A. Stiller-Reeve (mathew@stillerreeve.no)

**Abstract.** Do we miss something about «traditional" media such as handwritten letters and photography before the digital age? Some of the authors remember this age fondly, and we wanted to see if this fondness could be translated into a science dialogue project with school classes. We designed and carried out a communication process with 4 classes at different schools across Europe. During this process, each class would interact with a single scientist primarily via hand-written questions & letters, and a Polaroid photo album. The scientists would make this unique, one-of-a-kind album whilst on board a research expedition in the Barents Sea. We asked the question whether this process might show any benefits to the school students involved. To answer this, we asked the students to write up their thoughts on communicating with a scientist in this way. We analysed the texts and found that most students thought the letters and polaroid albums were a "beautiful experience". Others commented on how important it is to actually put pen to paper and write, since they use (almost) only digital media these days. Most importantly, the students learnt different elements of the science connected to the research expedition, but also about the scientific process in general. And, equally important, some of the students were surprised and thankful that the scientists took the time to communicate with them in such a personal way. These results could possibly have been achieved using other media, however the hand-written letters and Polaroids worked very well. They also seemed to conjure up some of the personal memories that we have about communication not so long ago. Maybe there is something to be said for slowing things down with our science communication projects and making them more personal and unique. This is something that snail-mail and making photo albums forces us to do.

## 1. INTRODUCTION

Are you old enough to remember the excitement of developing photographs? There was a time before smart phones and digital cameras, megapixels and insta-filters when photography was a stand-alone activity. Photography was mechanical. We





physically opened the cameras, inserted the film and wound it on. We waited patiently for the moment, memory or scene we

wanted to capture. As we clicked the button, the camera went through its mechanical actions. Photo taken, we wound on the film ready for the next. Until the film stopped winding on anymore. The film finished, we sent it away to be developed. We remember that little twang of anticipation when we received the developed photos. Our photos were rarely spectacular, but they invariably depicted happy and fond memories. And sometimes we slid these photographs into the envelopes together with a letter we had written by hand to family or friends.


Hand-written letters were another way of communicating that seems to be lost to time. We remember the thought that went into writing these letters, the stamps we stuck to the envelopes, and the post-boxes we slid them into. We remember waiting patiently for a reply. And we remember how exciting it was to hold the unopened letter in our hands, to tear open the envelope and to read the contents. Often several times.


Maybe only we, the authors, remember these ways of communicating so fondly. However, in these days of instant responses, emails, and unlimited cloud storage we wanted to see if reviving these "traditional" ways of communication could give a meaningful foundation to connect school classes with scientists.

Our project was certainly not the first to use hand-written letters and photography to connect science with a younger audience. We were highly inspired by "letters to a pre-scientist", which has been running for several years mainly in the USA (Madden, 2019). "Letters to a pre-scientist" connects individual school pupils with individual scientists and has had wonderful impact on the children and scientists taking part. They have seen that "interactions with a real scientist throughout the school year transform a scientist from a figure in a textbook into an actual person that the student knows and can aspire to emulate".

Hand-writing seems to be rapidly fading from education systems, something which several education researchers argue is likely detrimental (e.g. Karavanidou, 2017). Fortunati and Vincent (2014) found that writing/reading on paper is a "much more multi-sensorial experience than reading/writing on screen-keyboard", something which we hoped our project (much like "letters to a pre-scientist") would benefit from. In a way, we wanted to start a type of pen-pal correspondence between a scientist and a school which previous literature has shown to be very beneficial (Shandomo, 2009; Wiener and Matsumoto,

2014). However, we planned to combine these potential benefits of hand-written letters with the visual and personal aspect of traditional photography.

High-quality photography has potential to help science communication efforts (Zhu et al., 2021) engage people in conservation and biodiversity issues (Hanisch et al., 2019; Mittermeier, 2007), and even influence important political decisions

(Dunaway, 2006). Photography can connect people to ideas and each other. In our project, the scientists embark on a research expedition connected to a large geoscience research project (see below). On this expedition they would take polaroid photos and compile a photo album with hand-written description. The hope was that these albums would help make a meaningful



connection with a school class. High-quality photography was not a requirement in our project, but we hoped that the scientists' photographs would achieve some of the similar impacts on a smaller scale. We hoped that the scientists' photographs would

tell their research story and potentially increase engagement and interest amongst the school children. The personal story behind the photographs was what counted. Cooke et al. (2017) argue that through photography and video we can share so much about the research "journey". They state that "doing so can also help stakeholders understand the realities of science: things like uncertainty, variation, trial and error, and the surprising and surreal moments we all experience when we learn something new". Here, the stakeholders were school students in 3 different countries in mainland Europe/Scandinavia. But to

connect with these students, we needed to be sure we had an interesting research "journey" to communicate.

         Our "journey" was grounded in a project called Advancing Knowledge of Methane in the Arctic (AKMA). The AKMA project has been a collaborative project including scientists from the Arctic university of Norway in Tromsø, Norway and the Woods Hole Oceanographic Institution in Woods Hole, USA. The project aimed to develop a long term,

multidisciplinary education and research collaboration focused on Arctic methane sources, processes, ecosystems and geological history. One of the key objectives of the project was to provide exceptional training for the next generation of experts in Arctic marine sciences and greenhouse gas phenomena (https://akma-project.com/). Four of AKMA's early career scientists accepted the invitation to take part in this project that we called AKMA Polaroid. The communication between these scientists and the 4 school classes was (46 active students total) centred around the Arctic research cruise that happened in

May 2021. In other words, the whole communication process was designed around a real-life and real-time research expedition, which Pedrozo-Acuña et al. (2019) noted as beneficial to inspire "next generation geoscientists". There are several lovely examples of how scientists on expeditions can interact innovatively and imaginatively with school students to show them how science works and hopefully to broaden their career perspectives (e.g. Lebedev et al., 2019; Harrigan and Bower, 2019). We wanted to do that here but with the help of pen & paper, "snail-mail" and "traditional" photography.


         The aim of the AKMA Polaroid project was to develop a communication process where the scientists and school classes would communicate primarily via hand-written letters and polaroid photo albums made by the scientists during the research expedition. Throughout the process of the project (from development to execution) we kept asking ourselves the following research question: What kind of benefit do we see from using "traditional" communication media in a science

communication project?

## 2.   OUR PROCESS

### 2.1  The communication process with the schools

To answer our research question, we firstly needed to develop a communication process where photography (specifically

Polaroid photography) and handwritten letters were the main media of communication. We chose to use Polaroid photography





so that the scientists could receive the photographs immediately and compose a photo album whilst on the research expedition itself. The communication process comprised of seven main steps that we present in Figure 1. This process was developed with active feedback from the teachers to ensure relevancy for their students and their curriculum. Schools were invited from Norway, Italy and France mainly from within our existing networks and acquaintances. The 4 schools were all middle/high
schools with students between 15-17 years old. During our initial interactions with the teachers, we agreed that we would supply them with teaching materials that they could go through in their classes. These teaching materials would present some element of scientific knowledge and the communication process they would embark on with "their" scientist.

During the planning phase, we kept in mind that the teaching materials, and communication process as a whole,
should be useable by others. Others would likely find it challenging to reuse the materials if they focused on the AKMA science alone. So, instead we focused on a scientific research process. We would firstly describe a standard scientific process: from interest, to knowledge collection, to question forming, to research planning, to data collection and analysis, to communication. We then introduced the students to "their" scientists and explained how they would communicate. These teaching materials included a PowerPoint presentation that the teacher could present in class along with a video to help the teacher understand
what we were aiming to do. We hoped that the AKMA science would come to the foreground during the communication process between the scientist and the class.

Once the teachers had gone through the initial PowerPoint presentation with their class and introduced their scientist, then it was time to put pen to paper. The students were challenged to write down some questions inspired by what they had
just heard. These questions could be anything from general questions about why the scientists became scientists, to what exactly they will be doing on their research expedition. Figure 2 shows a selection of the questions the students posed. We see specific questions about the project, more general questions about science, and even personal questions about the scientists lives and why the scientists became scientists. To create a closer connection between student and the scientist, we could have tasked each student to send their hand-written questions to the scientists. However, we thought it best for the teacher to gather
the questions and send them to the scientist. Here is where the COVID-19 pandemic started to impact the process. Even though we challenged all the students to hand-write their questions, some of them could not deliver the questions to their teachers because the schools were under lock-down. Therefore, some of the teachers had to send the questions digitally in a Word document. This gave us the opportunity to notice a difference in the digital and hand-written letters. In the digital documents we only received (albeit very interesting) questions, whereas the hand-written documents included personal introductions, and
sometimes also hand-drawn pictures (see Figure 2).

The scientists received the questions and read through them carefully. They started to put together their responses. A couple of the scientists received over 50 questions, so they needed to pool some together and answer them at the same time. Others received around 20 questions so they could more easily answer individually. The scientists hand-wrote their responses



in, what turned out to be, rather lengthy, and personal letters. Some of these letters were over 10 pages long. Once the letters were sent in the post, it was time for the next exciting part of the project, where the scientist would balance scientific research and photo journalism.

      Before the research expedition on board the vessel Kronprins Håkon, each of the scientists received a Polaroid camera
(of their choice), 40 blank Polaroids, a blank photo album, glue, and gold/silver pens. Their task was to use the Polaroid camera to capture the science and the everyday life on board the ship. They should be inspired by the questions the classes had already asked to ensure that they shared stories about their research "journey" that the students would likely be interested in. The scientists also had to be careful with what they took pictures of. With only 40 negatives, each of them had to consider whether the scene was really one they wanted to capture. All of us in the project liked this aspect, as it made us feel the finiteness of
the resources we had at our disposal. With equipment in hand, the scientists were ready to go to sea.

      The AKMA research expedition happened between May 22 to June 9 2021. The vessel sailed from Longyearbyen on Svalbard and visited five sites characterized by seafloor methane and oil emissions (cold seepage), before docking in Tromsø. The scientists mapped seafloor morphology and collected sediment cores, rocks and fauna from the seabed using an
underwater robot -known as a remotely operated vehicle (ROV)- to study the effects of cold seepage on the surrounding marine habitats. During the expedition, each of the scientists took many unique Polaroid photos about all aspects of life onboard. They captured both the scientific and the personal aspects. They captured the excitement and the mundane. They captured the research instruments and the sports equipment. Figure 3 shows some examples of the pages in the photo albums that were made for the classes. Each album was a personal and unique mode of communication between each scientist and
"their" class.

      Once the scientists returned to shore, they were meant to post their albums to the classes. However, due to one of the scientist's travel plans, all the albums were delivered personally to the schools involved. Once received, the photo albums were circulated around the class. The students were tasked to think about some more questions inspired by the photos
and the descriptions.

      The final part of the interaction between each scientist and each class was a direct face-to-face link-up and discussion. These interactions were obviously influenced by the COVID-19 situation. Two of the link-ups were carried out online via Zoom. And two of the link-ups were carried out in person. Initially, these link-ups were meant to be the first time the scientists
and students met face-to-face. This was not the case since a couple of the classes had already linked-up with the scientists during the expedition itself. However, for one of the schools, this was the first face-to-face interaction. Here it is worth noting something (albeit anecdotally) important. Before the scientist arrived in person at the school, the students believed the scientist was just the teacher, who had devised an elaborate ruse to deliver teaching materials. They believed their teacher was playing



a trick on them. They were genuinely surprised when the scientist turned up and had spent the time writing to them and putting
together a photo album for them. Maybe this says something about the distance between science and society. It also reflects
how common R.O.V images and scientist talks through social media and mainstream media are. Consequently, feeding them
with in situ live images is not unusual at all, however being able to meet a scientist in person remains a lifetime experience
that can influence their career choices. Maybe this kind of personal and dedicated communication between scientists and
schools should be encouraged even more!


    Whether these final link-ups were in-situ or online, the students had many questions based on the photo albums and
the previous interactions during the AKMA Polaroid project. The scientists were asked about the technicalities of the research
equipment and whether they had discovered any scientific breakthroughs. They were asked about the overall goals of the
research and whether they achieved these goals. They were asked about their personal experience, how they dealt with potential
solitude on board, and whether they missed their family. They were also asked about general conditions in the Arctic, how
thick the ice is, and what animals one can see. Some asked about the basketball court onboard the vessel (take a close look at
Figure 3). These face-to-face interactions rounded off a two-way communication which had been dominated by hand-written
letters and personal and unique polaroid photo albums.

### 2.2 The evaluation

With the communication for the AKMA Polaroid project over, it was time to evaluate the process to see if the use of
these "traditional" ways of communication impacted the students taking part.

    The evaluation questions were designed to give us insight on the overall research question: What kind of benefit do
we see from using "traditional" communication media in a science communication project? Through discussion with the
project team- including input from the teachers involved- we formulated 3 intermediate questions that spoke to different
elements of the potential "benefits".

    We wanted to see how the students had experienced the interaction on a personal level. We wanted to know what
they had learnt and whether they had started to think differently about scientists in general. Since the number of evaluations
were likely to be rather low (we estimated 10-20 of the 46 pupils who had initially sent questions to the scientists), we decided
we would employ a narrative approach and let the students write freely. We would then analyze all the answers to see if any
clear themes percolated through. The questions were as follows (see supplementary materials for the full evaluation form the
students received which included a reminder of what had happened in the AKMA Polaroid project):

•    What did you think about using hand-written letters and polaroid photo albums? Could you write a text
        about what you feel about the communication with "your" scientist?



- Could you say something about what science you learnt through the interaction with the scientists using the letter and photo albums? (if you have not mentioned this already)?
- Could you say something about if this project has made you think differently about scientists in general (if you have not mentioned this already)?

We read through all the 17 evaluations that we received and applied a simple qualitative coding method (Saldaña, 2021) which we adapted to our study. We highlighted relevant and interesting quotes that contributed to answering our research question via the intermediate questions we posed to the students. Under each of the 3 intermediate questions we gathered these quotes into common insights. In this way our coding was deductive in nature since our intermediate questions were a starting point for our analysis. However, we also analysed the data to find common insights within the students' answers, and in this way we implemented a inductive approach, which let the data speak for itself (Linneberg and Korsgaard, 2019).

### 2.3  Ethical considerations

We carried out the evaluation according to guidelines from the Norwegian Agency for Shared Services in Education and Research (NSD) and those laid out by the British Educational Research Association (British Educational Research Association, 2018). Since we did *not* record any personal information of any kind during the evaluation, we did not require to formally notify NSD of the data collection. All students were informed about the evaluation by their teachers who acted as gatekeepers during this process. The students were considered of an age when they are "capable of forming their own views" and "should be granted the right to express those views freely" (BERA). The students were therefore asked if they *voluntarily* wanted to take part in the evaluations by their teachers. Since the survey was voluntary, we received considerably less answers than the total number who took part. All evaluations were anonymous and supplied via the teachers (and not directly from the students). To further ensure confidentiality and anonymity, the students' evaluations (from all schools) were then randomly ordered and temporarily stored for the analysis. Once the analysis was over and this article had passed the peer review process, the evaluations were deleted.

### 3.  THE EVALUATION RESULTS and DISCUSSION

In total, we received 17 evaluations of varying lengths. Some wrote long paragraphs and others wrote a short sentence or two. In this section we will go through the overarching insights that seemed to shine through in the 3 questions we posed the students.

### 3.1  Question 1: What did you think about using hand-written letters and polaroid photo albums?

We can start with whether the students thought the experience of communicating with these traditional media was positive or negative. The neutral and negative comments (of which 3 of 17 students came with) spoke to ways we could improve the



project, but also to wider issues around communication and education. Two of the students commented that they found it hard to "return to use handwritten letters" or "express my questions not using a PC". Maybe this says something more general about how students learn to communicate in schools these days. One of the other students commented directly on this issue and wrote that "it is important to write letters, because we are more careful when we are writing on paper than on screen". We received

one outright negative comment, which may also speak to wider issues. This student wrote that they enjoyed the project "in spite of the original PowerPoint, which was not really fascinating and captivating (without being mean, just objective)". Does this "objective" truth call for us to reconsider how we, as scientists, communicate with different audiences? Or does it say more about the project leader's ability to make "fascinating and captivating" PowerPoints? Indeed, Locritani et al. (2020) argued how using images in a fun way could engage more than a "normal frontal presentation". It's fully possible that the

Powerpoint was objectively boring. However, it could be that when weighed up against the "fun" polaroid and letter communication, then it certainly felt more boring. Whatever the reason, we were happy to hear that the traditional media was certainly preferred over using PowerPoint as a communication media.

This brings us to how the students responded positively to the use of hand-written letters and Polaroid albums. Overall,

16 of the 17 students responded with positive responses (note that some students gave both negative/neutral and positive comments). They said things like the hand-written letters and polaroid albums were "a beautiful experience", or that "this method is great and works wonderfully". This positive feedback also revealed another important impact of how we had designed the communication.

Some students (7 of 17) also commented the personal and reassuring connection they had experienced in communicating with "their" scientist. In forging out a communication process based on traditional media we hoped to make an inclusive, fun and accessible two-way dialogue between scientist and class as called for by Loroño-Leturiondo et al. (2019) (who discussed the subject of flooding and air pollution risk). In this way, we hoped to create a safe-space to exchange ideas about geoscience and for the students to ask *any* questions that they wanted. This element was nicely illustrated by a single

student who wrote that "by using handwritten letters and polaroid photos, it was easier to ask questions because it's less intimidating". Some of the others (3 of 17) commented specifically on how "thankful" they were that the scientists "took the time to write letters back to us". Others commented on the interaction felt "very personal" and that "using the hand-written letters and polaroid photo album made me feel like I was having a real interaction with "our" scientist". This speaks nicely to how we opened this article with our memories of how personal photography and letters used to feel. Maybe we lose some of

these personal connections by always communicating via computer and phone screens, short tweets and snappy emails. Maybe there is something to gain from slowing things down and taking the time to communicate meaningfully with a few. This is something which several of the students in this project apparently seemed to appreciate.



So overall, the students seemed very positive to the communication methods we had "tested" out on them, despite
work clearly needing on the initial PowerPoint presentation. We have to remember that a couple of the classes had also had
video link-ups with the scientists whilst on the expedition. This could have certainly influenced how they answered the
questions since the students would have gotten to know their scientist better because of these link-ups. We tried to avoid this
potential bias by getting the students to think only of the AKMA Polaroid process when they filled out the evaluation (see
Supplementary Material for how we conveyed this information).


We also needed to keep in mind that this was a *science* communication project. Despite the positive comments on the
communication media, we also needed to ensure that the students actually learnt about science through their experience.

### 3.2  Question 2: What science did you learn through the interaction with the scientists using the letter and photo albums?

We had designed the communication process in AKMA Polaroid to start with some general information about a scientific
process. After getting this information from the PowerPoint presentation, the students would get to know their scientist and
the science of the AKMA project through the communication process itself. In the evaluations from the students, we therefore
looked for whether the students thought they had learnt about both general scientific processes and the science of the project
itself.

In total, 6 of 15 students (only 15 of the students answered the second question) mentioned aspects of the AKMA
project itself. It was encouraging to see that they mentioned several different things such as "methane hydrates" and "their
impacts on the ocean", "fiery ice", coring, ROV-dives and methane bubbles. We certainly saw this in the discussions we had
with the schools during the final link-ups, where they also asked several detailed questions about how ROV's can resist high
pressures, how deep they can dive and what kind of tools it can carry. Just because not all students mentioned details associated
with AKMA science does not mean that they did not learn things about the project. It just means that they highlighted other
aspects of the science in their answers.


When it comes to more general science, 9 out of 15 highlighted this in their answers. The students wrote that they
learnt about the general "experience" of the scientists, and about how research actually takes place. One of the students
specifically mentioned that they learnt about the "missions, experiences, research, and financing". We found it encouraging
that the students were able to understand how broad the scientific process actually is and that many elements play a role in a
successful research project. Science is more than the "ability to parrot back what they are required to study", something which
often budding scientists do not understand before they start graduate studies (Volpe, 1984; Isaak and Hubert, 1999). A





successful research process also depends on good team work, which one of the students also began to understand when they wrote "I learned that there are a lot of crew members that work together".

### 3.3 Question 3: Did this project make you think differently about scientists in general?

We also wanted to see if this way of communicating between scientist and students made the students feel any differently about scientist or science as a career option. They had already commented that they were grateful for the time the scientists had used in communicating with them, but was there anything more?

Over half of the students (10 of the 16 who answered this question) stated that they had realized new things about scientists. Some of these new realisations focused on the "fun" aspects of the expedition. Some students mentioned the basketball court on board, but one also wrote that "it looks like a fun place to be on that ship in the arctic". A couple of the students also came to realise that a scientific career is not out of their reach. One student wrote that "before this project I saw scientific life as something a lot distant from myself, but through this project I realized that it is not that far away." Another student realized something many scientists do not realise themselves and wrote that "this project made me think that scientists can also do other jobs than what we know".

Only 3 of the 16 students did not think anything different about scientists through their experiences in AKMA Polaroid. However, all of these students had mentioned earlier that they had learnt new things and that they had appreciated being involved.

Finally, some of the students (3 of 16) commented specifically on how the project influenced their views on scientists as people. These comments centred around the passion that the scientists had shown during the communication process with one student saying that scientists do what they do "because they love it". However, one comment encompassed both the teamwork and the passion needed. This student wrote that they realized that "to be a scientist is a demanding job that requires determination and teamwork, but it's essential to improve our future and ensure a better future for the next generations". Obviously, we cannot directly link these sentiments to the use of hand-written letters and polaroid photo albums, but it is encouraging to hear such sentiments after the student had taken part in this project.

### 4. CONCLUDING REMARKS

The evaluations showed that many of the students clearly enjoyed connecting with scientists using hand-written letters, Polaroid photo albums, and a final face-to-face meeting. It is quite possible that this type of connection benefitted the students in several different ways, as we saw in the students' responses. We saw that they thought positively about the use of these



"traditional" media and conveyed that it was a "beautiful experience". They commented on the close connection they built with the scientist, and that they felt part of the expedition team. Not least, they learnt about the AKMA science and more general aspects about the scientific process.

We are aware that additional factors might have influenced students' feedback such as the live streaming from the ship, or whether the final link-ups were in-person or online. Despite this, our results are encouraging and show that the communication process we developed around these "traditional" media can have positive results. These positive results also have much to do with the welcoming and open way that the scientists communicated with the classes, and also the enthusiastic way that the teachers led the classroom activities and the interaction. But maybe it also has something to do with

time; by using letters and photo albums we slowed things down and we were forced to use more time in the communication process. In total, the classes and scientists interacted for around 6 months. This slow interaction could have led to a more personal connection, hence several students appreciating the time the scientists had used communicating with them.

    If a project like this was expanded in the future, it would be interesting to analyse the perspectives of the scientists

involved. If more scientists were involved, one could investigate what they take photos of and what aspects of research that they put emphasis on. It would also be interesting to interact with different classes in different ways so that one could more definitively say something about the impact of a specific media on the communication process.

    During the AKMA Polaroid project we certainly experienced how using "traditional" media could potentially make

a science communication project more personal and less intimidating. The process does not need costly technical solutions. It simply needs an initial connection with a class in a school, and a certain level of enthusism from the scientists and teachers involved. The students will unlikely remember this experience in the same way some of us authors remember letters and photography from earlier in life. However, the project shows that we can use traditional media to have a personal and meaningful (and fun!) communication with a few students that can also have a big impact.


## Acknowledgments:

We heartfully thank the teachers and students involved in this project. This work was supported by the Research Council of Norway through AKMA Advancing Knowledge on Methane in the Arctic (project number 287869) and CAGE (Center for

Excellence in Arctic Gas Hydrate Environment and Climate, project number 223259).



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

**Figure 1: The process we followed during the AKMA Polaroid project, from the initial project development project, through the communication activities, and ending with the evaluation questionnaire the students filled out.**




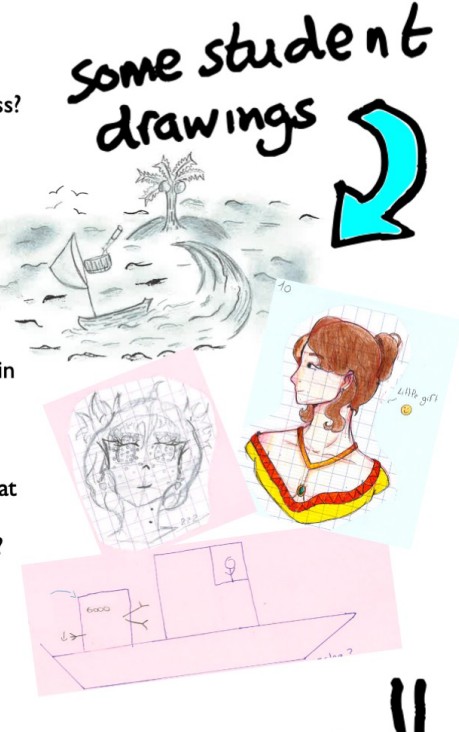

**Figure 2: Examples of some of the questions the students wrote to the scientists after their teachers had gone through what a research process looks like and introduced them to their scientist. Here are also some of the pictures the students included in in their hand-written questions.**






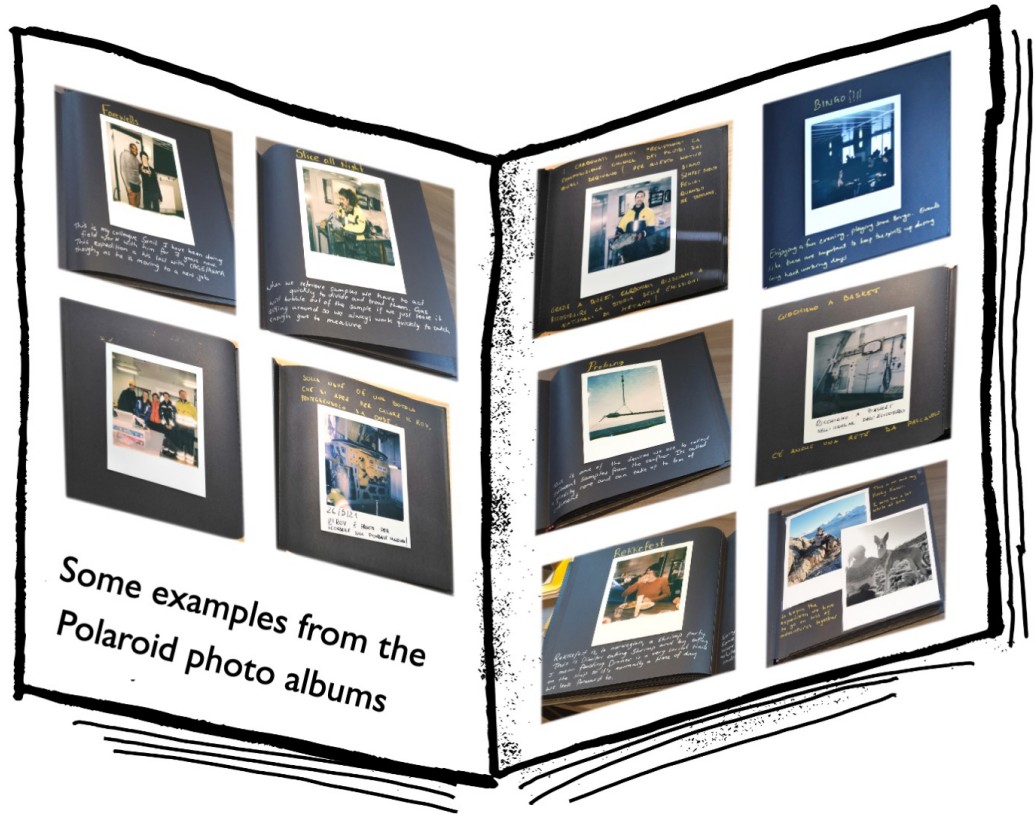

Figure 3: Some examples from the scientists' polaroid photo albums that they made for the classes they interacted with. The photos are simply meant to give an idea about how the albums were constructed. The captions are not meant to be readable in the present setting.
