# Peer review of "Hand-written letters and photo albums linking geoscientists with school classes"

_EGUsphere, 2022_

## Author Comment (AC1)

**RC1: Author response**

**RESPONSE: We thank the reviewer for taking the time to thoroughly review our article and present their views. We hope our responses are acceptable.**

Throughout the document, please check grammar and punctuation for consistency.   I have some considerations for the authors; these are listed below.

**RESPONSE:** Thank you. We will check the document again for grammar and punctuation.

1. Abstract: "Do we miss something about..." is more colloquial language since some readers might debate what is meant by "we."

**RESPONSE:** OK. We can make it more passive and write "Was something lost as society moved away from "traditional" media such as handwritten letters and photography and into the digital age?"

Also consider re-writing some parts of the manuscript and replace "we" with "the team that wrote this paper."

**RESPONSE:** This is a tricky one since we have been conscious about out use of "we". We understand that the all-encompassing "we" in the first sentence may confuse and we will deal with that. However, after that we feel it is quite clear that "we" is referring to the authors. In fact in some places, we write "we, the authors". We would need to reviewer to point out exactly where this issue is causing confusion for us to do anything about it.

2. Line 31: "Are you old enough to remember the excitement of developing photographs?"  This sentence appears to be unncessary and it is easiest to begin the paper with the sentence "There was a time before smart phones and digital... "

**RESPONSE:** We will make this change. Thanks

3. Line 41: What is meant by "lost to time"?  A newer generation of artists and photographers are still using film cameras.  The prevalence of film is slowly becoming a novelty and older methods of developing film are being deprecated.

**RESPONSE:** This is an interesting point and we are aware that film photography is becoming more popular again. We will refer to this in the updated version.

4. line 76: What is meant by "journey"?  Is this the conceptual process of engaging with students while engaged in a scientific investigation?

**RESPONSE:** In this sense it is similar to the research story we mention on line 70. However, since we refer to the work of Cooke et al (2017) we introduce their terminology of "journey". This term is used twice in sentences leading up to line 76, and we feel we introduce it clearly. The term "journey" is also the linking concept between the stress sentence of one paragraph and into the topic sentence of the next paragraph, where we describe the context of this "journey", i.e. the AKMA project.

5. line 105: How did the scientists engage and interact with the students?

**RESPONSE:** This question will need some expanding. In the line mentioned we state the "initial interactions with the *teachers"* and not the students. The proceeding paragraphs explain step-by-step how we engaged and interacted with the students themselves.

6. What is meant by "their" scientists?

**RESPONSE:** Thank you for bringing attention to this. Each class corresponded with a single scientist, which we refer to as "their" scientist. We will explain this more directly in the updated text.

7.  Is there some conceptual framework for engagement used to plan the interactions between the students and the scientists?

**RESPONSE:** No. Only an overarching research plan, where we had a clear research question and designed the engagement and evaluation around that. And, of course, we use the ethical frameworks that we refer to in the article.

8. line 206: Please add 1-2 sentences describing and providing background for the qualitative coding method. How was the method adapted?

**RESPONSE:** We describe the method in the sentences straight after line 207. Since this was not clear we will add ", in the following way" to the end of the sentence.

9. line 281: What is meant by "We had designed the communication process in AKMA Polaroid"?  The meaning is unclear, and please re-write.

**RESPONSE:** We will change this to "We designed the process in this project, AKMA Polaroid, to build upon general information about the scientific process, which the teachers presented in class. After this, the students would get to know their scientist and the science of the AKMA project through the iterative communication process of exchanging letter and photo albums."

10.  Can the authors briefly comment on the location of the schools?  Could any spatial differences in the responses be detected?

**RESPONSE:** This is a potentially interesting issue, but we feel it is not one that is within the scope of our study. This was a project to test out a method for communication not to detect differences across countries, cultures, economical standing, gender or a myriad of other potential issues. If we started analyzing one such issue then we would need to justify it, and where would that end? The question here was *if* this method could work, not what this method could tell us about other underlying societal issues.

---

## Author Comment (AC2)

**RC2: Author Response**

**RESPONSE: We thank the reviewer for taking the time to thoroughly review our article and present their views. The reviewer has come with excellent suggestions and feedback. With all respect for the time the reviewer has used on this review, we have disagreed with a couple of the suggestions that the reviewer has made. We hope our arguments come across politely. However, most of the suggestions will absolutely be dealt with in the revised manuscript.**

**General Comments**

This article reports an interesting work done by the authors in the field of geoscience education and is based on the educational approach through hand-written letters and photo albums. Moreover, the article highlights an important aspect of the science communication focusing on a more personal and engaging interaction between science communicators and audience. Today, the communication process is conditioned by digital technologies and this article faces and discus the possible contribution of an analogical and slower communication.

**RESPONSE:** Thank you for this evaluation. We are happy that you took this from the article.

Although the topic is relevant to the journal, the manuscript needs revision to improve the overall scientific level. In addition to the description the authors must indicate what is positive and what needs to be improved. For example, the activity you describe is a time-consuming activity, but not expensive.

**RESPONSE:** Thank you for this point. We can certainly add more points in the Discussion section (Concluding Remarks). However, we already have commented on several of the positives as the following examples show:

Line 335: *We saw that they thought positively about the use of these "traditional" media and conveyed that it was a "beautiful experience"*
Line 336: *They commented on the close connection they built with the scientist, and that they felt part of the expedition team.*
Line 337: *Not least, they learnt about the AKMA science and more general aspects about the scientific process.*

We also comment on some of the caveats, like the potential positive influence of the teacher's enthusiasm:

Line 343: *These positive results also have much to do with the welcoming and open way that the scientists communicated with the classes, and also the enthusiastic way that the teachers led the classroom activities and the interaction.*

And we comment on how time may have had a positive impact:

Line 344: *But maybe it also has something to do with time; by using letters and photo albums we slowed things down and we were forced to use more time in the communication process. In total, the classes and scientists interacted for around 6 months. This slow interaction could have led to a more personal connection, hence several students appreciating the time the scientists had used communicating with them.*

We also present ideas about things that could be improved, such as analysing the scientist's experience:

Line 349: *... it would be interesting to analyse the perspectives of the scientists involved.*

We also comment on the potential downside of using exactly the same process with all the classes and say that this could also be changed in the future:

Line 351: *It would also be interesting to interact with different classes in different ways so that one could more definitively say something about the impact of a specific media on the communication process.*

We would be happy to hear if the reviewer has other issues about "what is positive and what needs to be improved" that they suggest we should comment on. But at the moment, we feel we include statements on most of the issues the reviewer asks for.

See specific and technical comments.

**Specific Comments**

The article is generally well written but in some part it has repetitive phrases even not necessary for the reader's understanding. You may synthesize concepts that are too conversational, with some repetitions and non-essential information that make the reading difficult, especially in the introduction. Moreover, some phrases seem to be too emphatic for a scientific publication.

**RESPONSE:** Thank you for the feedback. However, without knowing which phrases the reviewer feels are too conversational of emphatic, we are unable to do much about this.

Considering you the 4 classes at different schools across Europe you should explain if the interaction between researchers and students was always in English. Did they write letters and comments in English? This is something that possibly may interact with their thoughts and the writing in a different language.

**RESPONSE:** This is certainly a point we can refer to in the text in a little more detail. We actually tried to let the students use the language they felt most comfortable using. However, some of the classes used English as a way to improve their language skills. We will include this in the revised version.

Generally, to make easier the reading, put quotes and citations at the end of the sentence and do not use brackets if not strictly necessary.

**RESPONSE:** The author will need to point out specifically which sentences he/she has an issue with here. We assume that the reviewer has indicated these in the numerous "technical corrections" below.

Regards the Evaluation and Cognitive assessments, you have very few data both respect to the total number and to the available sample. This needs to be discussed.

**RESPONSE:** The evaluation *must* be voluntary. This is important in any communication project and in particular with youth. The teachers promoted the evaluation, but we could not force it upon the students involved. We will comment on this further in the text. Since this was a project where we dealt with personal contact between scientists a school class, a simple multiple choice was not applied. We wanted to hear about the students actual experience and their feelings around the process. Hence we chose this more narrative approach. Even if just one student answered then their "story" would still be relevant and we would be able to extract useful feedback.

We saw clear themes across the students feedback. This shows that the corpus gives us some robust findings. However, we never comment on percentages, since we only have 17 (at most) answers to work from. We use actual numbers since this conveys the results more honestly.

Moreover, the first question contains more than one question and the evaluation process is not linear so you must take it into account analyzing the answers.

**RESPONSE:** The first question covers the overall communication process which included both handwritten letters and polaroid photo albums. These were the media that we were most interested in analyzing the use of, hence why we included them in one question. The students were free to discuss one or other or both.

We are unsure what the reviewer means by "not linear". We had three main issues we wanted to evaluate that were connected to our main research question. These three issues were relevant during the communication process from start to finish. These three main issues were 1). whether or not the "traditional" media elicited a positive experience, 2). whether the students learnt anything about the actual science, and 3). whether the experience made them think differently about scientists. Indeed all three of these elements would have been intermeshed and intertwined throughout the experience of the project. We could therefore not ask chronological (if that's what the reviewer means by linear) questions.

In the concluding remarks you must discuss and emphasize that you analyzes are based on little data and that perhaps by gaining more experience and adding more data your conclusions will become more robust.

**RESPONSE:** Ok, we will do that.

Finally a question: Have you observed gender-related aspects?

For example, did the class only interact with male scientists? Did girls and boys ask different questions? this is important for example when you write that some students have stated that they see the research profession as a real possibility.

**RESPONSE:** This is a potentially interesting issue, but we feel it is not one that is within the scope of our study. This was a project to test out a method for communication not to detect differences between gender or a myriad of other potential issues. For example, the other reviewer asked if we looked at differences across countries. If we started analyzing one such issue then we would need to justify it, and where would that end? The question here was *if* this method could work, not what this method could tell us about other underlying societal issues.

**Technical corrections**

remove

Maybe only we … foundly …

**RESPONSE:** This fond memory is what the whole project builds upon. Indeed it is emphatic, but it is true.

remove

Our project was certainly not the first to use hand-written letters and photography to connect science with a younger audience.

**RESPONSE:** We would need a reason why this sentence needs to be deleted. It is a topic sentence that introduces the paragraph to previous examples of similar communication projects. The second sentence then develops that idea. If the paragraph was *only* about the one example, then we certainly could delete the first sentence. However, the paragraph brings up several examples, and therefore requires a topic sentence that introduces the idea in a wider sense.

add

For our project we

**RESPONSE:** If we do not delete the sentence before (for the reason above) then we do not need to change the second sentence like this.

it would be interesting for discussion to understand why they consider it detrimental.

**RESPONSE:** This is because writing is a core skill. We will add a few words here.

remove the sentence in brackets

**RESPONSE:** OK

make a single sentence: remove the sentence in brackets and On this expedition they

**RESPONSE:** OK

replace a project called with the project

**RESPONSE:** OK

replace was (46 active students total) with  - 46 active students – was

**RESPONSE:** OK

replace which Pedrozo-Acunì a et al. (2019) noted as beneficial to inspire "next generation geoscientists"

with noted as beneficial to inspire "next generation geoscientists" by Pedrozo-AcunÌ a et al. (2019)

**RESPONSE:** OK

remove that we present

**RESPONSE:** OK

remove then it was time to put pen to paper

**RESPONSE:** We would need a stronger explanation why for removing this. A large part of this project is about actually putting pen to paper so we feel justified in writing this at some point during the story.

120-121 remove

These questions could be anything from general questions about why the scientists became scientists, to what exactly they will be doing on their research expedition.

**RESPONSE:** Again, we would need a stronger explanation why for removing this. This sentence communicates that the students had no strict rules on what they could ask. They could ask a very broad range of questions from "why the scientists became scientists, to what exactly they will be doing on their research expedition".

remove (albeit very interesting)

**RESPONSE:** OK

remove see

**RESPONSE:** OK

remove They started to put together their responses.

**RESPONSE:** OK

replace

Once the letters were sent in the post, it was time for the next exciting part of the project, where the scientist would balance scientific research and photo journalism.

with

In the next step of the project the scientist balanced the communication process with  scientific research and photo journalism.

**RESPONSE:**  OK

remove brackets

**RESPONSE:**  OK

remove

With equipment in hand, the scientists were ready to go to sea

**RESPONSE:**  There are two reasons this sentence is worded and positioned where it is. Firstly, it adds a little colour to the writing. Secondly, and most importantly, it links the description of the equipment (preceding paragraph) to the AKMA expedition itself (following paragraph). The idea "go to sea" transfers into the next paragraph which describes the expedition which was at sea.

replace

They captured both the scientific and the personal aspects. They captured the excitement and the mundane. They captured the research instruments and the sports equipment.

with

They captured both the scientific and the personal aspects, research instruments and excitement, sports equipment and the mundane.

**RESPONSE:**  These three sentences juxtapose seemingly opposite things. The science against the personal. The excitement against the boring/mundane and the science equipment against the sports equipment. They are meant to convey a sense that a research cruise is composed of many different, and often contrasting elements and experiences. We feel this is lost if we compile these comparative sentences together into one.

replace

Once the scientists returned to shore, they were meant to post their albums to the classes. However, due to one of the scientist's travel plans, all the albums were delivered personally to the schools involved.

with

Once the scientists returned to shore all the albums were delivered to the schools involved.

**RESPONSE:** We would like to keep this as we wrote it since it honestly conveys how our plans changed during the course of the project.

replace

Initially, these link-ups were meant to be the first time the scientists and students met face-to-face. This was not the case since a couple of the classes had already linked-up with the scientists during the expedition itself. However, for one of the schools, this was the first face-to-face interaction. Here it is worth noting something (albeit anecdotally) important.

with

For one of the schools, this was the first face-to-face interaction.

**RESPONSE:** Again, we would like to keep this as we wrote it since it honestly conveys how our plans changed during the course of the project.

explain:

R.O.V images

**RESPONSE:** OK

remove take a close look at

**RESPONSE:** Why? One needs to look closely at the photos in Figure 3 to see the basketball court, so why can we not convey this to the reader?

remove

With the communication for the AKMA Polaroid project over, it was time to evaluate the process to see if the use of these "traditional" ways of communication impacted the students taking part.

**RESPONSE:** OK

replace

Since the number of evaluations were likely to be rather low (we estimated 10-20 of the 46 pupils who had initially sent questions to the scientists), with

Since we estimated the number of evaluations to be rather low,

**RESPONSE:** OK

since the question contains more than one question, the process of analyzing the answers is not linear and you must take it into account when analyzing the answers

**RESPONSE:** We feel that we respond to this when we analyse it. The students comment on the general experience of using these "traditional" media and we highlight the comments where they focus on one or other of the media used. We would need some more explanation from the reviewer to understand what else we are missing.

only about 37% ? of students, please discuss this.

**RESPONSE:** OK. But we will add a comment about this further up in this section. We cannot be sure what aspect of this the reviewer would like us to comment on. However, 17 out of 46 possible responses is a good result for a completely voluntary survey.

add the acronym BERA

**RESPONSE:** Absolutely. This was a citation mistake.

Since the survey was voluntary, we received considerably less answers than the total number who took part.

Does that mean that they were obliged before? I suggest to remove the sentence

**RESPONSE:** Indeed they were obliged as the teacher deemed it part of their curriculum. This is a teacher's prerogative. However, we cannot analyse such interactions without abiding by ethical guidelines. In this sense the evaluation had to be voluntary. Hence we will keep this sentence as it is.

remove brackets

**RESPONSE:** OK

less than half of the initial sample of students discuss this and the meaning of analyzing so scarcely data.

**RESPONSE:** We are unsure what the reviewer means here. Each of the students who responded wrote sometimes lengthy feedback to us about their experience.

These stories give a wealth of data that we can analyze. We knew that not every student would answer (since it had to be voluntary), hence we chose a narrative approach that would give a wealth of insight from each *individual* student instead of clumping them together as a whole and extracting simple statistics.

remove can

**RESPONSE:**  OK

replace

The neutral and negative comments (of which 3 of 17 students came with) spoke to ways we could improve the project, but also to wider issues around communication and education with of the 17 comments were neutral and negative and indicate how we could improve the project, but also how widening issues around communication and education.

**RESPONSE:**  The suggested sentence does not communicate the same thing. Maybe if we change the "but" to "and" it will make sense:

*The neutral and negative comments (3 of 17) spoke to ways we could improve the project, and also to wider issues around communication and education.*

replace

One of the other students with

One student

 **RESPONSE:**  OK

remove brackets

**RESPONSE:**  OK

remove brackets

**RESPONSE:**  OK

remove

This speaks nicely to how we opened this article with our memories of how personal photography and letters used to feel.

**RESPONSE:**  Why? The point of this is to start closing the story arc from the way we framed the article in the beginning, and the elements that inspired the development of the project.

remove

We have to remember that

**RESPONSE:**  OK

replace

In total, 6 of 15 students (only 15 of the students answered the second question) mentioned aspects of the AKMA project itself.

with

Only 15 students answered the second question, 6 of which mentioned aspects of the AKMA project itself.

**RESPONSE:**  OK

replace

Over half of the students (10 of the 16 who answered this question) stated that they had realized new things about scientists with students answered this question, 10 of which stated that they had realized new things about scientists.

**RESPONSE:**  OK

replace

Finally, some of the students (3 of 16)

with

Finally, 3 out of 16 students

**RESPONSE:** OK

add that this could make up for the fact that our analysis has little data

**RESPONSE:** We would need more explanation here. How would analyzing the photographs make up for "little data" from the students. We realize that the reviewer believes that 17/46 students is not satisfactory. Again, we stress this is why we used a narrative approach. We are sure the reviewer understands that we could not force the students to answer. Hence, we had to design the survey to get as much information as possible from the students that wanted to answer. Given them the option to write freely gives the possibility.

Not mentioned in the text

**RESPONSE:** Yes it is. See Cooke et al. (2017). The reference actually starts on line 369.

Fig. 3 the images are not seen well

**RESPONSE:** We will take a close look at this.

---

## Author Response (AR2)

**Editor (minor revisions): Author response**

...thank you for thoroughly considering the reviewers' comments and revising your manuscript accordingly. However, I think you missed out on three critiques: Reviewer 1, comment 3, line 41 (original manuscript): What is meant by "lost to time"? You wanted to refer to the fact that "film photography is becoming more popular again", but I haven't found such reference.

**RESPONSE:** I'm sorry I did not pick up on this before! The paragraph transition now reads the following:

*"[...] Although "film photography has "recently witnessed a significant renaissance" (Marquardt and Andrae, 2022), it is still an activity many consider nostalgic.*

*Hand-written letters were another way of communicating that now seem increasingly lost to time. [...]"*

Reviewer 1, comment 6: "What is meant by "their" scientist"? => You wanted to explain this more directly, but as far as I can see haven't done so. Although I think it's clear enough what is meant by "their", please either explain, or send me a brief explanation why you changed your mind and don't do so.

**RESPONSE:** Even though this was obviously clear for some, it was not clear for all. Therefore, we added some text to present this more clearly. Now, we write, for example (line 116) *"We then introduced the students to the scientists they would communicate with and explained how they would communicate."*

Reviewer 2, technical comments, line 235 in the original manuscript: => Please change sentence to "The neutral and negative comments (3 of 17) spoke to ways we could improve the project, and also to wider issues around communication and education."

**RESPONSE:** Thank you for noticing this. We have made the requested change to the text.